# CO2: Consistent Contrast for Unsupervised Visual Representation Learning

**Chen Wei[1], Huiyu Wang[1], Wei Shen[2],[\*] Alan Yuille[1]**
[1]Johns Hopkins University    [2]Shanghai Jiao Tong University

## Abstract

Contrastive learning has been adopted as a core method for unsupervised visual representation learning. Without human annotation, the common practice is to perform an instance discrimination task: Given a query image crop, this task labels crops from the same image as positives, and crops from other randomly sampled images as negatives. An important limitation of this label assignment strategy is that it can not reflect the heterogeneous similarity between the query crop and each crop from other images, taking them as equally negative, while some of them may even belong to the same semantic class as the query. To address this issue, inspired by consistency regularization in semi-supervised learning on unlabeled data, we propose Consistent Contrast (CO2), which introduces a consistency regularization term into the current contrastive learning framework. Regarding the similarity of the query crop to each crop from other images as "unlabeled", the consistency term takes the corresponding similarity of a positive crop as a pseudo label, and encourages consistency between these two similarities. Empirically, CO2 improves Momentum Contrast (MoCo) by 2.9% top-1 accuracy on ImageNet linear protocol, 3.8% and 1.1% top-5 accuracy on 1% and 10% labeled semi-supervised settings. It also transfers to image classification, object detection, and semantic segmentation on PASCAL VOC. This shows that CO2 learns better visual representations for these downstream tasks.

## 1 Introduction

Unsupervised visual representation learning has attracted increasing research interests for it unlocks the potential of large-scale pre-training for vision models without human annotation. Most of recent works learn representations through one or more pretext tasks, in which labels are automatically generated from image data itself. Several early methods propose pretext tasks that explore the inherent structures within a single image. For example, by identifying spatial arrangement (Doersch et al., 2015), orientation (Gidaris et al., 2018), or chromatic channels (Zhang et al., 2016), models learn useful representations for downstream tasks. Recently, another line of works (Wu et al., 2018; Bachman et al., 2019; Hjelm et al., 2018; Tian et al., 2019; He et al., 2020; Misra & van der Maaten, 2020; Chen et al., 2020a), e.g. Momentum Contrast (MoCo), falls within the framework of contrastive learning (Hadsell et al., 2006), which directly learns relations of images as the pretext task. In practice, contrastive learning methods show better generalization in downstream tasks.

Although designed differently, most contrastive learning methods perform an instance discrimination task, i.e., contrasting between image instances. Specifically, given a query crop from one image, a positive sample is an image crop from the same image; negative samples are crops randomly sampled from other images in the training set. Thus, the label for instance discrimination is a one-hot encoding over the positive and negative samples. This objective is to bring together crops from the same image and keep away crops from different images in the feature space, forming an instance discrimination task.

However, the one-hot label used by instance discrimination might be problematic, since it takes all the crops from other images as equally negative, which cannot reflect the heterogeneous similarities between the query crop and each of them. For example, some "negative" samples are semantically similar to the query, or even belong to the same semantic class as the query. This is referred to as

---

[\*]corresponding author

"class collision" in Saunshi et al. (2019) and "sampling bias" in Chuang et al. (2020). The ignorance of the heterogeneous similarities between the query crop and the crops from other images can thus raise an obstacle for contrastive methods to learn a good representation. A recent work, supervised contrastive learning (Khosla et al., 2020), fixes this problem by using human annotated class labels and achieves strong classification performance. However, in unsupervised representation learning, the human annotated class labels are unavailable, and thus it is more challenging to capture the similarities between crops.

In this paper, we propose to view this instance discrimination task from the perspective of semi-supervised learning. The positive crop should be similar to the query for sure since they are from the same image, and thus can be viewed as labeled. On the contrary, the similarity between the query and each crop from other images is unknown, or unlabeled. With the viewpoint of semi-supervised learning, we introduce Consistent Contrast (CO2), a consistency regularization method which fits into current contrastive learning framework. Consistency regularization (Sajjadi et al., 2016) is at the core of many state-of-the-art semi-supervised learning algorithms (Xie et al., 2019; Berthelot et al., 2019b; Sohn et al., 2020). It generates pseudo labels for unlabeled data by relying on the assumption that a good model should output similar predictions on perturbed versions of the same image. Similarly, in unsupervised contrastive learning, since the query crop and the positive crop naturally form two perturbed versions of the same image, we encourage them to have consistent similarities to each crop from other images. Specifically, the similarity of the positive sample predicted by the model is taken as a pseudo label for that of the query crop.

Our model is trained with both the original instance discrimination loss term and the introduced consistency regularization term. The instance discrimination label and the pseudo similarity label jointly construct a virtual soft label on-the-fly, and the soft label further guides the model itself in a bootstrap manner. In this way, CO2 exploits the consistency assumption on unlabeled data, mitigates the "class collision" effect introduced by the one-hot labels, and results in a better visual representation. More importantly, our work brings a new perspective of unsupervised visual representation learning. It relaxes the stereotype that the pretext task can only be *self-supervised* which aims to construct artificial labels for every sample, e.g., a specific degree of rotation (Gidaris et al., 2018), a configuration of jigsaw puzzle (Noroozi & Favaro, 2016), and a one-hot label that indicates whether a crop comes from the same instance or not (Wu et al., 2018). In contrast, the pretext task can also be *self-semi-supervised*, allowing the task itself to be partially labeled. This relaxation is especially helpful when information for artificial label construction is not enough and imposing a label is harmful, such as the case of imposing the one-hot labels in instance discrimination.

This simple modification brings consistent gains on various evaluation protocols. We first benchmark CO2 on ImageNet (Deng et al., 2009) linear classification protocol. CO2 improves MoCo by 2.9% on top-1 accuracy. It also provides 3.8% and 1.1% top-5 accuracy gains under the semi-supervised setting on ImageNet with 1% and 10% labels respectively, showing the effectiveness of the introduced consistency regularization. We also evaluate the transfer ability of the learned representations on three different downstream tasks: image classification, object detection and semantic segmentation. CO2 models consistently surpass their MoCo counterparts, showing that CO2 can improve the generalization ability of learned representation. Besides, our experiments on ImageNet-100 (Tian et al., 2019) demonstrate the efficacy of CO2 on SimCLR (Chen et al., 2020a), showing the generality of our method on different contrastive learning frameworks.

## 2 METHOD

In this section, we begin by formulating current unsupervised contrastive learning as an instance discrimination task. Then, we propose our consistency regularization term which addresses the ignorance of the heterogeneous similarity between the query crop and each crop of other images in the instance discrimination task.

### 2.1 CONTRASTIVE LEARNING

Contrastive learning (Hadsell et al., 2006) is recently adopted as an objective for unsupervised learning of visual representations. Its goal is to find a parametric function $f_\theta : \mathbb{R}^D \to \mathbb{R}^d$ that maps an input vector $\mathbf{x}$ to a feature vector $f_\theta(\mathbf{x}) \in \mathbb{R}^d$ with $D \gg d$, such that a simple distance measure (e.g., cosine distance) in the low-dimensional feature space can reflect complex similarities in the high-dimensional input space.

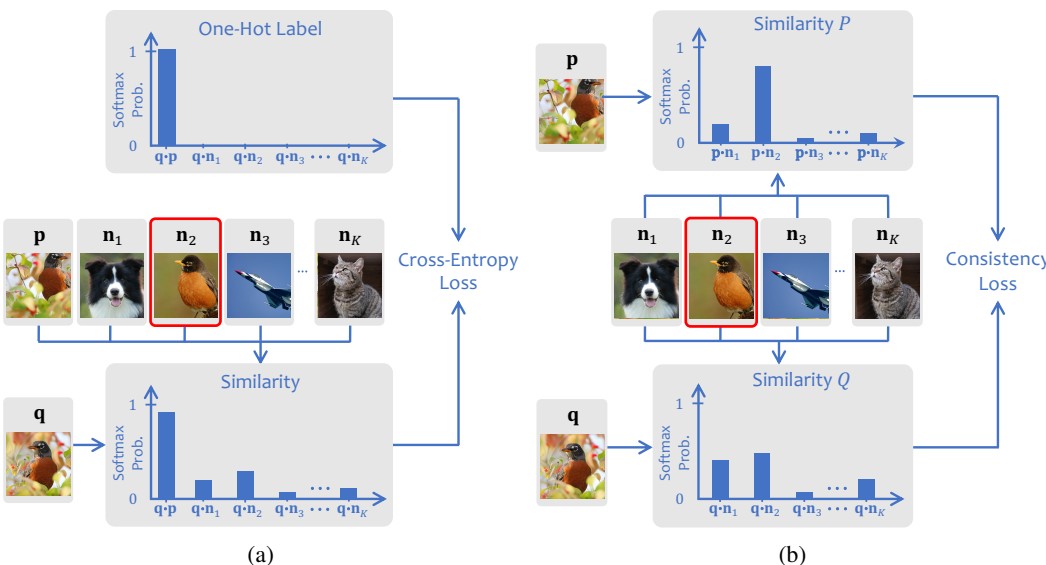

Figure 1: Illustration of (**a**) instance discrimination and (**b**) our consistency regularization term. $\mathbf{q}$ is a query and $\mathbf{p}$ is a positive key, both are encoded from crops of the same image. $\{\mathbf{n}_k\}_{k=1}^K$ are negative keys, encoded from random crops. In (**a**), the similarities are softmax cosine distances between $\mathbf{q}$ and each key. ($\mathbf{p}$ and $\{\mathbf{n}_k\}_{k=1}^K$). These similarities are optimized towards an artificial one-hot label which identifies $\mathbf{p}$ among all keys. However, some negatives can be semantically similar but not reflected by the one-hot label (e.g., the one rounded by a red box). In (**b**), our proposed consistency regularization encourages the agreement between $P$, the positive-negative similarities, and $Q$, the query-negative similarities, reflecting the heterogeneous similarities between the query/positive and the negatives.

For each input vector $\mathbf{x}_i$ in the training set $\mathbb{S}$, the similarity measure in the input space is defined by a subset of training vectors $\mathbb{S}_i \subset \mathbb{S}$, called similarity set. The sample $\mathbf{x}_i$ is deemed similar to samples in the similarity set $\mathbb{S}_i$, but dissimilar to samples in $\mathbb{S} \setminus \mathbb{S}_i$. Then, the contrastive objective encourages $f_\theta(\mathbf{x}_j)$ to be close to $f_\theta(\mathbf{x}_i)$ in the feature space if $\mathbf{x}_j \in \mathbb{S}_i$, and otherwise to be distant.

By training with contrastive loss, the similarities defined by the similarity set determine characteristics of the learned representation and the mapping function $f_\theta$. For example, if the similarity is defined as samples from the same semantic class, then $f_\theta$ will probably learn invariances to other factors, e.g., object deformation. In the supervised setting, this definition of similarity requires a large amount of human labeling. On the contrary, unsupervised contrastive learning exploits similarities with no need of human labels. One natural definition of unsupervised similarity is multiple views of an image, as explored by many recent methods. For example, random augmented crops (Wu et al., 2018; Ye et al., 2019; He et al., 2020; Chen et al., 2020a;b) of an image could be defined as a similarity set. In this case, the contrastive objective is effectively solving an instance discrimination task (Wu et al., 2018) as illustrated in Figure 1a.

The training of this instance discriminator involves randomly sampling a query crop $\mathbf{x}^q \in \mathbb{S}_i$, a positive crop $\mathbf{x}^p \in \mathbb{S}_i$ from the same image, and $K$ negative crops $\{\mathbf{x}^k \in \mathbb{S} \setminus \mathbb{S}_i\}_{k=1}^K$ from other images. These $K + 2$ crops (the query, the positive, and $K$ negatives) are encoded with $f_\theta$ respectively, $\mathbf{q} = f_\theta(\mathbf{x}^q), \mathbf{p} = f_\theta(\mathbf{x}^p), \mathbf{n}_k = f_\theta(\mathbf{x}^k)$. Then, an effective contrastive loss function, InfoNCE (Hjelm et al., 2018), is written as:

$$\mathcal{L}_{ins} = -\log \frac{\exp(\mathbf{q} \cdot \mathbf{p}/\tau_{ins})}{\exp(\mathbf{q} \cdot \mathbf{p}/\tau_{ins}) + \sum_{k=1}^K \exp(\mathbf{q} \cdot \mathbf{n}_k/\tau_{ins})}, \tag{1}$$

where $\tau_{ins}$ is a temperature hyper-parameter (Hinton et al., 2015). This loss can be interpreted as a cross entropy loss that trains the model to discriminate the positive crop (labeled as 1) from negative crops (labeled as 0) given the query crop. We denote this loss as $\mathcal{L}_{ins}$ as it performs an instance discrimination task. One direct instantiation of InfoNCE loss, represented by SimCLR (Chen et al., 2020a), formulates $f_\theta$ as an end-to-end encoder. In this case, two crops of the same image are exchangeable or symmetric to each other as both are encoded by $f_\theta$. The final loss is also symmetric

with either one of the two crops as the query and the other crop as the positive. Another popular instantiation, represented by MoCo (He et al., 2020), encodes the query with $f_\theta$ and encodes the positive and the negatives with $f_{\theta'}$ which is the moving average of $f_\theta$. In this case, only $\mathbf{q}$ can propagate gradients, which causes $\mathcal{L}_{ins}$ to be asymmetric.

## 2.2 CONSISTENT CONTRAST

The one-hot labels used by InfoNCE loss is effective, showing good generalization ability across tasks and datasets (Chen et al., 2020b;a). Nevertheless, we argue that the hard, zero-one labels is uninformative. Specifically, crops from other images are taken as equally negative as they are all labeled as 0. This is contradictory to the fact that some so-called "negative" crops can be similar or even in the same semantic class, especially when $K$ is large. For example, SimCLR (Chen et al., 2020a) uses 16,382 negative samples in a batch, and MoCo (He et al., 2020; Chen et al., 2020b) uses a memory bank of 65,536 features as negative samples. Even worse, the current objective forces negatives to be as far from the query as possible, with larger weights for closer negatives since they are "hard negatives". However, these "hard negative" crops in fact tend to be semantically close. These issues impair good representation learning because the one-hot labels can not faithfully reflect the heterogeneous similarities between the query crop and the crops from other images.

Although generating labels based on instance discrimination is trivial, revealing the similarity between two arbitrary crops is exactly what we want to learn from unsupervised pre-training. Therefore, the label of the similarity between the query crop and each crop from other images is of little hope to get. This situation is similar to the usage of unlabeled data in semi-supervised learning setting, in which consistency regularization is widely used to propagate knowledge from labeled data to discover the structures in unlabeled data. Inspired by this, we propose to encourage the consistency between the similarities of crops from the same image, i.e., the query crop and the positive crop. We illustrate the consistency regularization in Figure 1b.

First, we denote the similarity between the query $\mathbf{q}$ and the negatives $\mathbf{n}_i (i \in \{1, \ldots, K\})$ as:

$$Q(i) = \frac{\exp(\mathbf{q} \cdot \mathbf{n}_i / \tau_{con})}{\sum_{k=1}^{K} \exp(\mathbf{q} \cdot \mathbf{n}_k / \tau_{con})} \, , \tag{2}$$

where $\tau_{con}$ is also a temperature hyper-parameter. $Q(i)$ is the probability that the query $\mathbf{q}$ selects $\mathbf{n}_i$ as its match from $\{\mathbf{n}_k\}_{k=1}^{K}$. Similarly, the similarity between the positive $\mathbf{p}$ and the negatives is written as:

$$P(i) = \frac{\exp(\mathbf{p} \cdot \mathbf{n}_i / \tau_{con})}{\sum_{k=1}^{K} \exp(\mathbf{p} \cdot \mathbf{n}_k / \tau_{con})} \, . \tag{3}$$

We impose the consistency between the probability distributions $P$ and $Q$ by using symmetric Kullback-Leibler (KL) Divergence as the measure of disagreement:

$$\mathcal{L}_{con} = \frac{1}{2} D_{\mathrm{KL}}(P \| Q) + \frac{1}{2} D_{\mathrm{KL}}(Q \| P) \, . \tag{4}$$

When $\mathbf{p}$ and $\mathbf{q}$ are encoded by the same end-to-end encoder $f_\theta$, it is natural to use symmetric KL as their disagreement measure, since $\mathbf{p}$ and $\mathbf{q}$ are exchangeable. Even when $\mathbf{p}$ and $\mathbf{n}_i$ are encoded by the momentum encoder $f_\theta'$, symmetric KL empirically works as well as forward KL, i.e., $D_{\mathrm{KL}}(P \| Q)$, as shown in Section 3.5. Thus, we use symmetric KL as a unified objective for both cases.

The total loss is a weighted average of the original instance discrimination loss term and the consistency regularization term:

$$\mathcal{L} = \mathcal{L}_{ins} + \alpha \mathcal{L}_{con} \, , \tag{5}$$

where $\alpha$ denotes the coefficient to balance the two terms. It is possible to merge the two terms by creating a unique label containing information of both the one-hot label and the pseudo similarity label, but we find the weighted average can already get good performance and is easy to control.

The pseudo label is informative to reveal the similarity between the query $\mathbf{q}$ and each $\mathbf{n}_i$, while the one-hot label is unable to provide such information, since it only describe co-occurrence within one image. Note that, the pseudo label is also dynamic since the embedding function $f_\theta$ is updated in every training step, and thus generating better pseudo labels during training. It indicates that the unsupervised embedding function and the soft similarity labels give positive feedback to each other.

Table 1: Linear classification protocol on ImageNet-1K

| Pretext Task | Arch. | Head | #epochs | Top-1 Acc. (%) |
|---|---|---|---|---|
| ImageNet Classification | R50 | - | 90 | 76.5 |
| Exemplar (Dosovitskiy et al., 2014) | R50w3$\times$ | - | 35 | 46.0 |
| Relative Position (Doersch et al., 2015) | R50w2$\times$ | - | 35 | 51.4 |
| Rotation (Gidaris et al., 2018) | Rv50w4$\times$ | - | 35 | 55.4 |
| Jigsaw (Noroozi & Favaro, 2016) | R50 | - | 90 | 45.7 |
| *Methods based on contrastive learning*: | | | | |
| InsDisc (Wu et al., 2018) | R50 | Linear | 200 | 54.0 |
| Local Agg. (Zhuang et al., 2019) | R50 | Linear | 200 | 58.2 |
| CPC v2 (Hénaff et al., 2019) | R170$_w$ | - | ~200 | 65.9 |
| CMC (Tian et al., 2019) | R50 | Liner | 240 | 60.0 |
| AMDIM (Bachman et al., 2019) | AMDIM$_{large}$ | - | 150 | 68.1 |
| PIRL (Misra & van der Maaten, 2020) | R50 | Linear | 800 | 63.6 |
| SimCLR (Chen et al., 2020a) | R50 | MLP | 1000 | 69.3 |
| MoCo (He et al., 2020) | R50 | Linear | 200 | 60.6 |
| MoCo (He et al., 2020) + CO2 | R50 | Linear | 200 | 63.5 |
| MoCo v2 (Chen et al., 2020b) | R50 | MLP | 200 | 67.5 |
| MoCo v2 (Chen et al., 2020b) + CO2 | R50 | MLP | 200 | 68.0 |

Table 2: Top-5 accuracy for semi-supervised learning on ImageNet

| Pretext Task | 1% labels | 10% labels |
|---|---|---|
| Supervised Baseline | 48.4 | 80.4 |
| InsDisc (Wu et al., 2018) | 39.2 | 77.4 |
| PIRL (Misra & van der Maaten, 2020) | 57.2 | 83.8 |
| MoCo (He et al., 2020) | 62.4 | 84.1 |
| MoCo (He et al., 2020) + CO2 | 66.2 | 85.2 |
| MoCo v2 (Chen et al., 2020b) | 69.5 | 85.1 |
| MoCo v2 (Chen et al., 2020b) + CO2 | 70.6 | 85.4 |

Our method is simple and low-cost. It captures the similarity to each $\mathbf{n}_i$ while introducing unnoticeable computational overhead with only one extra loss term computed. This is unlike clustering based unsupervised learning methods, which are costly, since they explicitly compute the similarity sets in the training set after every training epoch (Caron et al., 2018; Zhuang et al., 2019; Li et al., 2020; Caron et al., 2020).

## 3 EXPERIMENTS

Herein, we first report our implementation details and benchmark the learned representations on ImageNet. Next, we examine how the unsupervised pre-trained models transfer to other datasets and tasks. We then analyze the characteristics of our proposed method.

### 3.1 LINEAR CLASSIFICATION

**Setup** We mainly evaluate CO2 based on MoCo (He et al., 2020) and MoCo v2 (Chen et al., 2020b). Both of them use instance discrimination as pretext task, while MoCo v2 adopts more sophisticated design choices on projection head architecture, learning rate schedule and data augmentation strategy. We test CO2 on MoCo for its representativeness and simplicity. On MoCo v2, we evaluate how CO2 is compatible with advanced design choices. We also demonstrate the impact of CO2 on the end-to-end contrastive framework in Section 3.5.

The unsupervised training is performed on the train split of ImageNet-1K (Deng et al., 2009) without using label information. We keep aligned every detail with our baseline MoCo to effectively pinpoint the contribution of our approach, except the number of GPUs (MoCo uses 8 GPUs while we use 4). A further search on MoCo-related hyper-parameters might lead to better results of our

Table 3: Transfer learning performance on PASCAL VOC datasets

| Pretext Task | Image Classification | Object Detection | | | Semantic Segmentation |
|---|---|---|---|---|---|
| | mAP | $AP_{50}$ | $AP_{all}$ | $AP_{75}$ | mIoU |
| ImageNet Classification | 88.0 | 81.3 | 53.5 | 58.8 | 74.4 |
| Rotation (Gidaris et al., 2018) | 63.9 | 72.5 | 46.3 | 49.3 | - |
| Jigsaw (Noroozi & Favaro, 2016) | 64.5 | 75.1 | 48.9 | 52.9 | - |
| InsDisc (Wu et al., 2018) | 76.6 | 79.1 | 52.3 | 56.9 | - |
| PIRL (Misra & van der Maaten, 2020) | 81.1 | 80.7 | 54.0 | 59.7 | - |
| SimCLR (Chen et al., 2020a)* | - | 81.8 | 55.5 | 61.4 | - |
| BYOL (Grill et al., 2020)* | - | 81.4 | 55.3 | 61.1 | - |
| SwAV (Caron et al., 2020)* | - | 81.5 | 55.4 | 61.4 | - |
| SimSiam (Chen & He, 2020)* | - | 82.4 | 57.0 | 63.7 | - |
| MoCo (He et al., 2020) | - | 81.5 | 55.9 | 62.6 | 72.5 |
| MoCo (He et al., 2020) (*our impl.*) | 79.7 | 81.6 | 56.2 | 62.4 | 72.6 |
| MoCo (He et al., 2020) + CO2 | 82.6 | 81.9 | 56.0 | 62.6 | 73.3 |
| MoCo v2 (Chen et al., 2020b) | 85.0 | 82.4 | 57.0 | 63.6 | 74.2 |
| MoCo v2 (Chen et al., 2020b) + CO2 | 85.2 | 82.7 | 57.2 | 64.1 | 74.7 |

*\* Results reported in Chen & He (2020).*

method. For the hyper-parameters of CO2, we set $\tau_{con}$ as 0.04, $\alpha$ as 10 for MoCo-based CO2, and $\tau_{con}$ as 0.05, $\alpha$ as 0.3 for MoCo v2-based CO2. Please refer to the appendix for more detailed implementation description.

## 3.2 LINEAR CLASSIFICATION

We first benchmark the learned representations on the common linear classification protocol. After the unsupervised pre-training stage, we freeze the backbone network including the batch normalization parameters, and train a linear classifier consisting of a fully-connected layer and a softmax layer on the 2048-D features following the global average pooling layer. Table 1 summaries the single-crop top-1 classification accuracy on the validation set of ImageNet-1K. Our method consistently improves by 2.9% on MoCo and by 0.5% on MoCo v2. We also list several top-performing methods in the table for reference. These results indicate that the representation is more linearly separable on ImageNet with consistency regularization, since the consistency regularization mitigates the "class collision" effect caused by semantically similar negative samples.

## 3.3 SEMI-SUPERVISED LEARNING

We next perform semi-supervised learning on ImageNet to evaluate the effectiveness of the pre-trained network in data-efficient settings. Following (Wu et al., 2018; Misra & van der Maaten, 2020; Chen et al., 2020a), we finetune the whole pre-trained networks with only 1% and 10% labels which are sampled in a class-balanced way. Table 2 summaries the mean of the top-5 accuracy on the validation set of ImageNet-1K over three runs. The results for MoCo and MoCo v2 are produced by us using their officially released models. The proposed consistency regularization term can provide 3.8% and 1.1% top-5 accuracy gains for MoCo with 1% and 10% labels respectively. CO2 also improves from MoCo v2 by 1.1% top-5 accuracy with 1% labels, and by 0.3% with 10% labels.

## 3.4 TRANSFER LEARNING

To further investigate the generalization ability of our models across different datasets and tasks, we evaluate the transfer learning performance on PASCAL VOC (Everingham et al., 2015) with three typical visual recognition tasks, i.e., image classification, object detection and semantic segmentation. Table 3 reports the transfer learning performance comparing with other methods using ResNet-50. CO2 shows competitive or better performance comparing with the corresponding baselines, In addition, it achieves better performance than state-of-the-art unsupervised representation learning methods.

**Image Classification** Following the evaluation setup in Goyal et al. (2019), we train a linear SVM (Boser et al., 1992) on the frozen 2048-D features extracted after the global average pool-

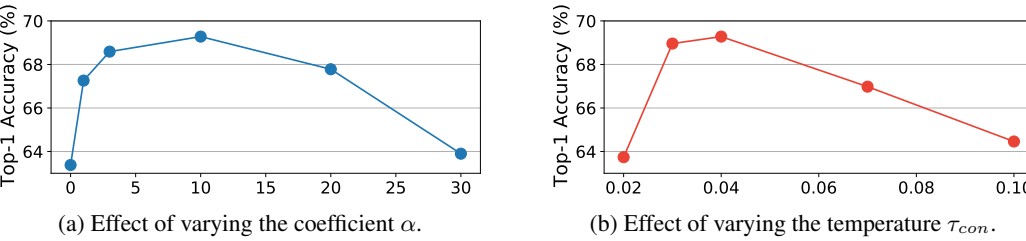

(a) Effect of varying the coefficient $\alpha$.  (b) Effect of varying the temperature $\tau_{con}$.

Figure 2: Ablation on the effect of hyper-parameters.

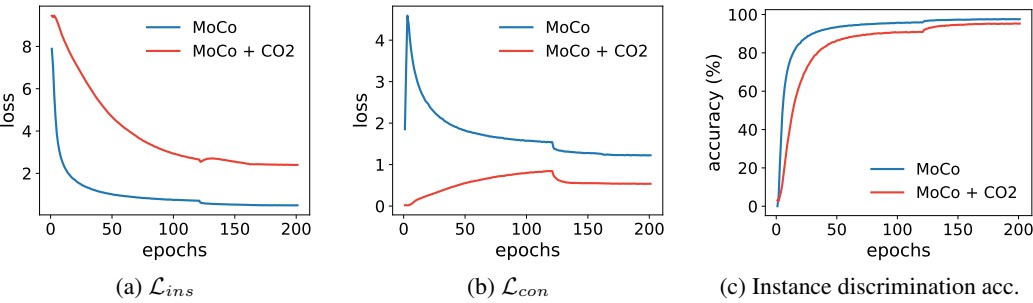

(a) $\mathcal{L}_{ins}$  (b) $\mathcal{L}_{con}$  (c) Instance discrimination acc.

Figure 3: Training curves of ResNet-18 on ImageNet-100.

ing layer. The results of MoCo are produced by us with their official models. In this case, CO2 is 2.9% better than MoCo, and 0.2% than MoCo v2.

**Object Detection** Following the detection benchmark set up in He et al. (2020), we use Faster R-CNN (Ren et al., 2015) object detector and ResNet-50 C4 (He et al., 2017) backbone, and all the layers are finetuned including the batch normalization parameters. The numbers of our method are averaged over three runs. Our reproduced results for MoCo are also listed in the table for reference. CO2 provides 0.3% $AP_{50}$ gains on both MoCo and MoCo v2.

**Semantic Segmentation** We follow the settings in He et al. (2020) for semantic segmentation. Results are average over three runs. Similarly, we include our reproduced results of MoCo as a reference. The result of MoCo v2 is produced by us using its officially released model. CO2 gives 0.9% mIoU improvement upon MoCo, and 0.5% upon MoCo v2, which finally surpasses its supervised counterpart.

The overall transfer learning improvements, though consistent, are smaller than linear classification and semi-supervised learning. Similar observations have also been made in Chen et al. (2020b). We hypothesize that the current unsupervised contrastive methods, which bring close different crops from the same image, reduce the representation's sensitivity to location which is useful for tasks like detection. It is still an open question which properties of an unsupervised representation benefit the transfer ability to various downstream tasks.

## 3.5 ANALYSIS

In this section, we study the characteristics of the proposed method on a smaller backbone ResNet-18 and a smaller dataset ImageNet-100 due to the consideration of the computational resource. ImageNet-100 is firstly used in Tian et al. (2019) and consists of 100 randomly selected classes from all $1,000$ classes of ImageNet.

**Hyper-parameter** Our method introduces two new hyper-parameters, the coefficient of consistency regularization term $\alpha$, and its temperature $\tau_{con}$. In Figure 2, we show the top-1 accuracy of a linear classifier on models pre-trained by CO2 with different hyper-parameters. In Figure 2a, we fix the temperature $\tau_{con}$ as 0.04 and vary the coefficient $\alpha$. The best coefficient is 10. We see that by using the consistency regularization term, the linear classification accuracy can be boosted from 63.6% to 69.2%. Increasing $\alpha$ to 20 and beyond causes performance degeneration. We hypothesize that the model is over-regularized by the consistency loss, and thus it loses some discrimination among different instances. In Figure 2b, we fix the coefficient to be 10 and varying the temperature. As other consistency regularization methods (e.g., Berthelot et al. (2019b)), temperature $\tau_{con}$ effectively influences the quality of the learned representation, and the best to use is 0.04.

**Training Curves**  In Figure 3 we show the training curves of the instance discrimination loss $\mathcal{L}_{ins}$, the consistency loss $\mathcal{L}_{con}$ and the instance discrimination accuracy. Instance discrimination accuracy represents the percent of query crops which successfully select their corresponding positive crops, i.e., successfully identify their instances. MoCo is trained with $\mathcal{L}_{ins}$ only and its $\mathcal{L}_{con}$ is just calculated out for comparison. We see that $\mathcal{L}_{ins}$ of MoCo drops quickly from the beginning at the cost of a jump of $\mathcal{L}_{con}$. As the training proceeds, $\mathcal{L}_{con}$ of MoCo decreases spontaneously, possibly because more semantic knowledge has been learned, but it is still relatively high. Training with $\mathcal{L}_{con}$ and $\mathcal{L}_{ins}$ together, i.e., MoCo + CO2, $\mathcal{L}_{con}$ is kept very low from beginning, and $\mathcal{L}_{con}$ increases gradually since the model is trained to discriminate between images at the same time. At the end of the training, $\mathcal{L}_{con}$ stays much lower than $\mathcal{L}_{con}$ of MoCo.

We also notice that with CO2, the instance discrimination accuracy drops from 97.57% to 95.26%. Although CO2 results in lower instance discrimination accuracy, it still does better in the downstream classification task. The linear classification accuracy improves from 63.6% to 69.2%, as shown in Figure 2a. It suggests again that there is a gap between instance discrimination and the downstream tasks.

**Comparison with Label Smoothing**  With the consistency regularization term, our approach assigns soft pseudo labels to crops from other images. This looks similar to label smoothing regularization on supervised classification (Szegedy et al., 2016), a useful trick which assigns a small constant value to the labels of all the negative classes to avoid overconfidence. We equip MoCo with label smoothing, i.e., assigning a small constant value to crops from other images (the "negatives"). Surprisingly, it reports 61.2% linear classification accuracy, 2.4% lower than MoCo alone. This suggests that assigning a constant value as label smoothing can be harmful for unsupervised contrastive learning, since it ignores the heterogeneous similarity relationship. And it is better to assign labels according to the similarities as our consistency regularization.

**End-to-End Encoder**  To further verify the effectiveness of the proposed consistency regularization term on different contrastive learning frameworks, we apply CO2 to Sim-CLR (Chen et al., 2020a), a representative method with an end-to-end encoder (without a momentum encoder). The results are presented in Table 4. On ImageNet-100 (Tian et al., 2019) with a ResNet-18, SimCLR obtains 68.9% top-1 linear classification accuracy with batch size 128 and temperature $\tau_{ins}$ 0.1. Equipped with CO2 whose coefficient $\alpha$ is 0.07 and temperature $\tau_{con}$ is 1.0, the linear classification accuracy is boosted to 72.3%. The improvement demonstrates that CO2 can be applied to different unsupervised contrastive frameworks and improve the quality of the learned representation regardless of whether using a momentum encoder or not.

Table 4: Linear classification accuracy using an end-to-end encoder and with different choices of $\mathcal{L}_{con}$. The results are summarized as mean and standard deviation over three different runs.

| Method | Acc. (%) |
| --- | --- |
| SimCLR | $68.9_{\pm 0.06}$ |
| SimCLR + CO2 | $72.3_{\pm 0.14}$ |
| MoCo | $63.1_{\pm 0.29}$ |
| MoCo + Forward KL | $69.6_{\pm 0.27}$ |
| MoCo + Reverse KL | $65.1_{\pm 0.52}$ |
| MoCo + CO2 | $69.7_{\pm 0.41}$ |

**Varying the choices of $\mathcal{L}_{con}$**  We ablate on different variants of $\mathcal{L}_{con}$ (Eq. 4) on MoCo including forward KL ($D_{\mathrm{KL}}(P\|Q)$), reverse KL ($D_{\mathrm{KL}}(Q\|P)$), and the objective of CO2, i.e., symmetric KL. Each of models uses a coefficient $\alpha$ of 10 and a temperature $\tau_{con}$ of 0.04. We present the linear classification accuracy in Table 4. Our CO2 (symmetric KL) improves over the baseline MoCo by a large margin, from 63.1% to 69.7%. Forward KL alone improves similarly to 69.6%. And reserve KL alone can also provide a nontrivial 2.0% gain in accuracy.

## 4  RELATED WORK

Our method falls in the area of unsupervised visual representation learning, especially for image data. In this section, we first revisit various design strategies of pretext tasks for unsupervised learning. Then we elaborate on the pretext tasks based on contrastive learning, which is the focus of our work. Next, we review the methods using consistency regularization in semi-supervised learning, which inspire our work.

**Unsupervised Learning and Pretext Tasks**  To learn from unlabeled image data, a wide range of pretext tasks have been established. The models can be taught to specify the relative position of a patch (Doersch et al., 2015), solve spatial jigsaw puzzles (Noroozi & Favaro, 2016; Wei et al.,

2019), colorize gray scale images (Zhang et al., 2016; Larsson et al., 2017), inpaint images (Pathak et al., 2016), count objects (Noroozi et al., 2017), discriminate orientation (Gidaris et al., 2018), iteratively cluster (Caron et al., 2018; Zhuang et al., 2019; Asano et al., 2019; Zhong et al., 2020), generate images (Donahue et al., 2016; Donahue & Simonyan, 2019), *etc*. Doersch & Zisserman (2017) evaluates the combination of different pretext tasks. Kolesnikov et al. (2019) and Goyal et al. (2019) revisit and benchmark different pretext tasks.

**Contrastive Learning** Contrastive learning (Hadsell et al., 2006) recently puts a new perspective on the design of pretext task and holds the key to most state-of-the-art methods. Most of them perform an instance discrimination task while differ in i) the strategies to synthesize positives and negatives, and ii) the mechanisms to manage a large amount of negatives. The synthesizing can base on context with patches (Hjelm et al., 2018; 2019), random resized crops with data augmentation (Wu et al., 2018; Ye et al., 2019; Bachman et al., 2019; He et al., 2020; Chen et al., 2020a), jigsaw puzzle transformation (Misra & van der Maaten, 2020) or luminance-chrominance decomposition (Tian et al., 2019). Regarding the mechanisms to maintain negative features, some methods (Wu et al., 2018; Misra & van der Maaten, 2020) keep tracking the features of all images, some directly utilize the samples within the minibatch (Chen et al., 2020a; Tian et al., 2019; Ye et al., 2019), and He et al. (2020) proposes to use a momentum encoder. Grill et al. (2020) recently proposes to only use positive examples without negatives. Recently, Li et al. (2020) argues that the lack of semantic structure is one fundamental weakness of instance discrimination, and proposes to generate prototypes by k-means clustering. However, the computational overhead and the degeneration introduced by clustering are difficult to address. Chuang et al. (2020) also points out the possible sampling bias of instance discrimination, and proposes a debiased objective.

**Consistency Regularization** Consistency regularization is an important component of many successful semi-supervised learning methods. It is firstly proposed in Sajjadi et al. (2016), encouraging similar predictions on perturbed versions of the same image. Besides the consistency regularization on unlabeled data, the model is simultaneously trained with a supervised loss on a small set of labeled data. Several works made improvements on the way of perturbation, including using an adversarial transformation (Miyato et al., 2018), using the prediction of a moving average or previous model (Tarvainen & Valpola, 2017; Laine & Aila, 2017), and using strong data augmentation (Xie et al., 2019). Recently, several larger pipelines are proposed (Berthelot et al., 2019b;a; Sohn et al., 2020), in which consistency regularization still serves as a core component.

The instance discrimination loss in unsupervised contrastive learning is analogous to the supervised loss in semi-supervised learning, as both of them rely on some concrete information, i.e., co-occurrence in one image and human annotation, respectively. Meanwhile, CO2 on the similarities between crops is analogous to consistency regularization on unlabeled samples of semi-supervised methods as their labels are both unknown. The main difference, however, is that semi-supervised methods crucially rely on the supervised loss to warm up the model, while there is no human annotation at all in unsupervised contrastive learning. Our work presents an example that a model learned completely without human annotations can also generate surprisingly effective pseudo labels for similarities between different crops and benefit from consistency regularization.

## 5 DISCUSSION

Unsupervised visual representation learning has shown encouraging progress recently, thanks to the introduction of instance discrimination and the contrastive learning framework. However, in this paper, we point out that instance discrimination is ignorant of the heterogeneous similarities between image crops. We address this issue with a consistency regularization term on the similarities between crops, inspired by semi-supervised learning methods which impose consistency regularization on unlabeled data. In such a simple way, the proposed CO2 consistently improves on supervised and semi-supervised image classification. It also transfers to other datasets and downstream tasks.

More broadly, we encourage researchers to rethink label correctness in existing pretext tasks. Taking instance discrimination as an example, we show that a pretext task itself could be, in fact, a semi-supervised learning task. It might be harmful to think of the pretext task as a simple pure supervised task by assuming the unknown labels are negatives. In addition, our work relaxes the stereotype restriction that pretext task labels should always be known and clean. We hope this relaxation can give rise to novel pretext tasks which exploit noisy labels or partially-available labels, making a better usage of the data without human annotation.

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

# A    APPENDIX

## A.1    IMPLEMENTATION DETAILS OF CONTRASTIVE PRE-TRAINING

We evaluate our approach based on MoCo (He et al., 2020). MoCo has two different encoders to encode queries and keys respectively. The query encoder is updated with respect to the loss function, while the key encoder is an exponential moving average of the query encoder. The keys are stored in a dynamic memory bank, whose entries are updated at every training step with the current mini-batch enqueued and the oldest mini-batch dequeued. The backbone is a standard ResNet-50 (He et al., 2016), and features after the global average pooling layer are projected to 128-D vectors (Wu et al., 2018), normalized by $\ell_2$ norm. The size of the memory bank (i.e., the number of negative samples) is 65,536 and the momentum to update the key encoder is 0.999. $\tau_{ins}$ is 0.07 for MoCo variants and 0.2 for MoCo v2 variants, which are the default settings of these two methods.

We use momentum SGD with momentum 0.9 and weight decay 1e-4. The batch size is 256 on 4 GPUs. To prevent potential information leak with Batch Normalization (BN) (Ioffe & Szegedy, 2015), shuffling BN (He et al., 2020) is performed. The model is trained for 200 epochs with the initial learning rate of 0.03. The learning rate is multiplied by 0.1 after 120 and 160 epochs for MoCo v1, while cosine decayed (Loshchilov & Hutter, 2016) for MoCo v2. We keep aligned all training details with MoCo except the number of GPUs. This could be problematic since it changes the per-worker minibatch size, which is related to potential information leaks pointed by He et al. (2020). However, we do not notice much difference when reproducing MoCo with 4 GPUs. Our reproduced MoCo v2 with 4 GPUs reaches the accuracy of 67.6% on the linear classification protocol, 0.1% higher than 67.5% reported in its paper. For the hyper-parameters of the proposed consistency term, we set $\tau_{con}s$ as 0.04 and $\alpha$ as 10 for the MoCo v1-based CO2, and $\tau_{con}$ as 0.05, $\alpha$ as 0.3 for the MoCo v2-based variant.

## A.2    IMPLEMETATION DETAILS OF DOWNSTREAM TASKS

**Linear Classification**    We freeze the backbone network including the batch normalization parameters, and train a linear classifier consisting of a fully-connected layer followed by softmax on the 2048-D features following the global average pooling layer. We train for 100 epochs. The learning rate is initialized as 15 and decayed by 0.1 every 20 epoch after the first 60 epochs. We set weight decay as 0 and momentum as 0.9. Only random cropping with random horizontal flipping is used as data augmentation.

**Semi-Supervised Learning**    We finetune the pre-trained model for 20 epochs with learning rate starting from 0.01 for the base model and 1.0 for the randomly initialized classification head, decayed by 0.2 after 12 and 16 epochs. Momentum is set to 0.9. Weight decay is 5e-4 for MoCo v1 and 1e-4 for MoCo v2. Only random cropping with random horizontal flipping is used as data augmentation.

**Classification on PASCAL VOC**    Following the evaluation setup in Goyal et al. (2019), we train a linearSVM (Boser et al., 1992) on the frozen 2048-D features extracted after the global average pooling layer. The models are trained on `trainval2007` split and tested on `test2007`. The hyper-parameters are selected based on a held-out subset of the training set.

**Detection on PASCAL VOC**    Following the detection benchmark set up in He et al. (2020), we use FasterR-CNN (Ren et al., 2015) object detector and ResNet-50 C4 (He et al., 2017) backbone, implemented in Detectron2 (Wu et al., 2019). We finetune all the layers including the batch normalization parameters for 24k iterations on the `trainval07+12` split and test on `test2007` set. The hyper-parameters are the same as the counterpart with supervised ImageNet initialization and MoCo. To calibrate the small feature magnitude due to the output normalization in the unsupervised pre-training stage, two extra batch normalization layers are introduced, one is followed by the regional proposal head whose gradients are divided by 10 and the other is followed by the box prediction head.

**Segmentation on PASCAL VOC**    Following the setup in He et al. (2020), an FCN-based (Long et al., 2015) architecture with atrous convolutions (Chen et al., 2017) is used and ResNet-50 is the backbone. The training set is `train_aug2012` (Hariharan et al., 2011) and the testing set is `val2012`. Initialized with CO2 models, we finetune all layers for 50 epochs ( 33k iterations) with batch size 16, initial learning rate 0.003, weight decay 1e-4 and momentum 0.9.

