# OpenReview forum: "CO2: Consistent Contrast for Unsupervised Visual Representation Learning"
_ICLR.cc/2021/Conference — ICLR 2021 Poster_

### Official Review · AnonReviewer4 · 2020-10-22
**Good Submission with Some Concerns**

**Rating:** 6
**Confidence:** 5

**Review:**

##########################################################################

Summary:

This paper proposed a consistency regularization for unsupervised visual representation learning. This paper argues that the instance discrimination task performed by most contrastive learning methods merely uses one-hot labels, which cannot reflect the similarities between the query sample and negative samples. In order to tackle this problem, this paper proposed a consistency regularization method to generate pseudo labels and encourages the query sample and its positive sample to have consistent similarities to negative samples. The proposed consistency regularization method is simple and low-cost compared to clustering-based methods.


##########################################################################

Reasons for score:

Overall, I vote for acceptance, but some concerns on implementations lower my score. The idea of consistency regularization is simple and low-cost, but achieves significant improvement on the popular MoCo baseline. My main concerns are that (1) the improvement on MoCo v2 is much smaller than MoCo while the weight of the consistency regularization term is small (10 for MoCo, 0.3 for MoCo v2), (2) the usage of symmetric KD divergence, (3) the discussion about Mean Teacher [1]. I will change my rating depending on the feedback.


##########################################################################

Pros:

P1: The research problem, contrastive learning for unsupervised visual representation learning, is popular and important in the CV community.

P2: The proposed consistent contrast (CO2) method is simple but effective. Compared to clustering-based methods, CO2 is less time-consuming.

P3: The performance of linear classification, semi-supervised learning, transfer learning is reported.

P4: The analysis with a smaller backbone and dataset is provided as an ablation study.

##########################################################################

Cons:

C1: Section 3.1 states that CO2 can be easily applied to other contrastive learning mechanisms. It is acceptable to me that CO2 is evaluated with MoCo. However, it is not convincing that the application of other methods is easy. The reason is that MoCo and SimCLR have different a design of key encoders. CO2 works well with the momentum updated encoder. However, there is no evidence that CO2 also works well with the end-to-end encoder, since the query and key encoders are updated simultaneously. I doubt that this conclusion is misleading and overstating.

C2: In Section 3.1, $\alpha$ is set as 10 for MoCo while 0.3 for MoCo v2. The improvement of CO2 over MoCo is 2.9% while only 0.5% for MoCo v2. It is not clear why $\alpha$ is set at different levels and why the improvement over MoCo v2 is not significant.

C3: This paper is closed to Mean Teacher [Tarvainen & Valpola, 2017]. The contrastive loss (instance discrimination) is like the classification cost in Mean Teacher, while the symmetric KL Diverfebce (consistency regularization) is like the consistency cost. The comparison, especially the difference, between CO2 and Mean Teacher should be discussed.

##########################################################################

Questions during the rebuttal period:

Q1: It is not clear to me why a symmetric KD divergence (i.e., D(P|Q)+D(Q|P)) is used in Eq. (4) rather than asymmetric KD divergence (i.e., D(Q|P)). In MoCo, the encoder for computing keys is momentum-updated rather than updated by back-propagation. Therefore, Eq.(4) becomes

$\mathcal{L}_{con}=\frac{1}{2}\sum^K_i-P(i)\log Q(i)+Q(i)\log Q(i)-Q(i)\log P(i)$

Since the gradient of $P(i)$ is stopped, $P(i)\log P(i)$ is ignored. If the gradient of $Q$ in $D_{KL}(Q||P)$ is stopped, then $D_{KL}(Q||P)$ does not contribute to back-propagation. If not, $Q(i)\log Q(i)$ works like a regularization term as minimizing $Q(i)\log Q(i)$ may encourage a sharp $Q(i)$. I wonder whether the sharpness implicitly boosts the performance. I hope that more explanation about the symmetric KD divergence, especially (1) why "symmetric" rather than only $D_{KL}(Q||P)$. If "symmetric" is necessary, additional ablations and references/citations will help. (2) Whether $Q(i)\log Q(i)$ is minimized. If so, more explanation is expected.

Q2: In section 2.2, Eq. (2) and (3) only consider the similarity between the query and negatives. I wonder why the similarity between the query and positive is ignored. Although the latter is used in Eq.(1), I wonder whether there is any difference if the similarity between the query and positive is also considered in Eq. (2) and (3).

#########################################################################

Suggestions:

S1: In Page 3, this paper states that $q$, $p$ and $n_k$ shares the same encoder $f_\theta$. However, $q$, $p$ and $n_k$ are obtained with different encoders using MoCo. This part could be carefully revised to be more general.

#########################################################################

References:

[1] Antti Tarvainen and Harri Valpola. Mean teachers are better role models: Weight-averaged consistency targets improve semi-supervised deep learning results. In NeurIPS, 2017.






------------------------------------------------------------------------------

Thank the authors for their good job during rebuttal.

Most of my questions have been addressed. However, my main concern is still that the improvement of CO2 on MoCo v2 is incremental. During the rebuttal stage, the authors did not include more empirical evidence on whether CO2 can improve MoCo v2 well, or theoretical analysis on why the improvement is incremental, both of which are accepteable to me. This severely limits the contribution of this paper. In all, I would not change my score.

---

> ### Author Response · Authors · 2020-11-19
> **Response to Reviewer 4**
>
> We thank Reviewer 4  for the valuable comments. Below we discuss the points you have raised in detail.
>
> **C1: No evidence that CO2 also works well with the end-to-end encoder.**
>
> We agree MoCo and SimCLR have different designs of key encoders. We demonstrate the effectiveness of CO2 with end-to-end encoders in **G3** in our general response.
>
> **C2: Why $\alpha$ is set at different levels and why the improvement over MoCo v2 is not significant.**
>
> - The $\alpha$ for MoCo and MoCo v2 are not directly comparable: MoCo and MoCo v2 differ in the use of MLP and data augmentation, and thus require different hyper-parameters (e.g. they use different temperatures, 0.07 and 0.2), so we also select our $\alpha$ for both versions. Besides, the different temperatures of MoCo and MoCo v2 affect the scales of gradients significantly [1], which further affects the choice of $\alpha$.
> - As discussed in **G4**, we select our hyper-parameters on ImageNet-100, so it is also possible that a larger alpha leads to better performance for MoCo v2.
>
> **C3: The comparison, especially the difference, between CO2 and Mean Teacher should be discussed.**
>
> Thank you for bringing up this great analogy. In fact, we believe this analogy can be applied between CO2 and other semi-supervised learning methods (eg., [2]) which leverage the consistency to exploit unlabeled data. This is exactly our motivation to introduce consistency into contrastive learning: we take the similarity between the query and the positive as *labeled* as they are from the same image, and the similarity between the query/positive and the negatives as *unlabeled*.
>
> The difference is two-fold. First, Mean Teacher emphasizes the benefit of generating pseudo-labels with the momentum encoder, while we show CO2 can work with and without the momentum encoder. Second, Mean Teacher, and other semi-supervised learning methods, crucially rely on the classification cost on labeled samples to get an initial model, while CO2 does not involve any human annotations but solely rely on the unsupervised contrastive loss. Moreover, Mean Teacher applies the classification cost and the consistency cost on two distinct sets of samples, i.e., a label set and an unlabeled set, respectively. While our work applies two costs on the same set of unlabeled samples which are all we have.
>
> **Q1: Why a symmetric KL divergence is used in Eq. (4).**
>
> Thanks for the question and the reformulation of Eq. (4)!
> - “Symmetric” is not necessary. Please see **G2** in our general response.
> - $Q(i)\log Q(i)$ is minimized, i.e. the entropy of $Q(i)$ is maximized, which encourages a smooth $Q(i)$. However, this entropy term is not critical. As shown in **G2**, forward KL (i.e. $-P(i)\log Q(i)$ alone) can already work on par with symmetric KL. In addition, we ablate using the entropy term alone without tuning loss weights, and we get a 38.8% linear classifier accuracy since the total loss is overwhelmed by the entropy term. These two results confirm that it is the consistency that makes CO2 work, not the entropy.
>
> **Q2: Why the similarity between the query and positive is ignored.**
>
> Thanks for this insightful question.
> - As you pointed out, the similarity of the query and the positive has been addressed in Eq. (1).
> - Conceptually, we believe considering $q\cdot p$ in Eq. (2) and Eq. (3) is in line with our motivation for consistency regularization. Technically, since Eq. (1) draws the query and the positive together and repels the negatives away, $q\cdot p$ is usually much larger than any of $q\cdot n_i$. If taking $q\cdot p$ into Eq. (2) and Eq. (3), it could overwhelm other terms, making them contribute less to the gradients. However, it is the similarity between the query/positive and the negative samples that is of our interest in CO2.
>
> **S1: $q$, $p$, $n_k$ shares the same encoder $f_{\theta}$.**
>
> Thanks for the suggestions and we are sorry for the misleading. As in **G3** in our general response, we present our idea under the general framework of end-to-end contrastive learning which uses the same encoder $f_{\theta}$, and we take MoCo with a momentum encoder as an approximation for large negative sample size. We have shown that CO2 can help with and without the momentum encoder. We will clarify the relationship between CO2 and the end-to-end/momentum encoder in the updated draft.
>
>
> We would be more than happy to discuss any further questions!
>
>
> [1] Geoffrey Hinton, Oriol Vinyals, and Jeff Dean. Distilling the knowledge in a neural network. In *NeurIPS Deep Learning and Representation Learning Workshop*, 2015.
>
> [2] Kihyuk Sohn, David Berthelot, Chun-Liang Li, Zizhao Zhang, Nicholas Carlini, Ekin D Cubuk, Alex Kurakin, Han Zhang, and Colin Raffel. FixMatch: Simplifying semi-supervised learning with consistency and confidence. In *NeurIPS*, 2020.

---

### Official Review · AnonReviewer2 · 2020-10-27
**A new loss function for smoothing the pseudo labels in unsupervised contrastive loss**

**Rating:** 7
**Confidence:** 3

**Review:**

*Summary*
This paper proposes an extension, coined as CO2, to InfoNCE contrastive loss used in semi/unsupervised methods. CO2 is based on the premise that the query-negative crop similarity distribution and positive-negative crop similarity distribution should be alike. The proposed method yields significant improvements for the linear classification protocol using ImageNet while the improvements for downstream transfer learning tasks such as object detection is marginal. Aside from the performance improvements, the paradigm where the researchers think of the pretext task as a downstream task and improve the pseudo labels by using pseudo-pseudo labels is very interesting. Proposed CO2 method provides a relatively simple way to achieve this.


Authors build upon the intuition that among the many crops the algorithm uses as negatives, it is highly likely that at least a few positive samples exist. These unknown “positive” crops should yield a high similarity to the query, but this is not possible to enforce as these crops’ labels are, by definition, unknown. Instead the authors suggest that positive-negative similarities and query-negative similarities should be alike. A KL divergence term (between positive-negative similarity distribution and query-negative distribution) is used to implement this constraint.

*Novelty*
To the best of my knowledge this is a novel approach. It is also surprising that using pseudo-pseudo labels to correct the wrong assumptions of pseudo labels is working reasonably well.

*Impact*
I believe this paper will have a significant impact. Accuracy improvements on downstream tasks are diminished when the authors use MoCo-v2, suggesting that their method may not always yield significant benefits. However, aside from the numeric accuracy improvements on the downstream tasks, the proposed idea is very simple, seems easy to incorporate in other methods and likely opens new research directions.

*Clarity*
The paper is well written and easy to follow. It is also generally clear but for some of my questions/comments please see below.

*Evaluation*
Authors use their loss function with MoCo and MoCo-v2 and report relatively small improvements over MoCo-v2. It is an open question whether the proposed loss function would result in large performance gains for other methods.
On the transfer learning side, authors report marginal improvements for image classification, object detection and semantic segmentation tasks. It is not possible to form an opinion on how statistically significant these improvements are. Still, the authors provide a comparison with label smoothing which implies that CO2 is a beneficial addition.

*Strengths (Reasons to accept)*
This is a relatively simple loss function extension, applicable to other methods.
Experiments are implying that the method works well and improves upon the state of the art (under reasonable resource constraints).
As pointed out in the discussion, the paradigm of relaxing the pretext task’s label constraints (in a way, a learned label smoothing) is likely to open new research directions.

*Weaknesses (Reasons to reject)*
Transfer learning improvements are marginal (and in the object detection case CO2 results in an unexplained 0.2% drop in AP). I would expect to see a discussion about the reasons behind the discrepancy between the large improvements for semi-supervised learning (or linear classification) vs. the marginal improvements for transfer learning.
Only the object detection and semantic segmentation task experiments have been repeated (3 times). The rest of the experiments are, I believe, single runs. This makes the reader question the significance of the reported results.
A more in depth explanation for the choice of the particular loss in Eq. 4  would benefit the reader. Why use symmetric loss? Please see my questions below.

*Questions and other comments to the authors*
In Figure 1 I would like to see where the labels (both One-hot and Pseudo) come from, at least in the caption. The same is true for the similarity graphs, and the authors should consider adding axis-labels and named ticks. Finally, increasing the arrowhead sizes would make everything easier to follow.
Even though we can not expect the authors to reproduce 1000 epoch and >4000 batch-size methods common to recent semi-supervised techniques, I would still like to see the impact of CO2 using a vanilla ResNet backbone (without the momentum encoder).
I believe in Eq. 4, the first term alone should be enough to ensure that the “learned” query extractor distribution (Q) matches the “ground truth” distribution (P). In the next few paragraphs authors state P to be dynamic, but with just the first term P would be dynamic as both Q and P depend on the same network. As such, it is not clear to me why the authors chose a symmetric divergence.
Finally, authors should refrain from using the letter P both for the distribution and for the pseudo label. This is an unnecessary overloading of notation.

---

> ### Author Response · Authors · 2020-11-19
> **Response to Reviewer 2**
>
> We thank Reviewer 2 for the valuable comments. Below we discuss the points you have raised in detail.
>
> **Q1: Transfer learning improvements are marginal.**
>
> Thank you for suggesting this insightful discussion. We agree the transfer learning improvements, especially on object detection, is marginal compared with those of semi-supervised learning and linear classification. Actually, similar observations have been made in other works. In MoCo v2 [1], using MLP instead of aug+ provides 2.8% more linear accuracy, but 0.2% drop in AP50, 0.4% drop in AP and 0.75% drop in AP75. This suggests that “linear classification accuracy is not monotonically related to transfer performance in detection”, quoted from MoCo v2. We hypothesize one reason could be current contrastive learning frameworks, which bring close different crops from the same image, reduce the representation’s sensitivity of location which is useful for tasks like detection. It is still an open question which properties of an unsupervised representation are important for the transfer ability to various downstream tasks such as detection, and we leave it to future work.
>
> **Q2: The rest of the experiments are, I believe, single runs. This makes the reader question the significance of the reported results.**
>
> We agree that more runs for experiments can better demonstrate the improvements. We plan to provide more runs for results in Table 2, the semi-supervised learning results.
>
> **Q3: It is not clear to me why the authors chose a symmetric divergence.**
>
> We are sorry for the confusion. Please see **G2** about our choice of symmetric KL divergence.
>
> **Q4: Figure 1 can be improved.**
>
> Thanks for the suggestions. We will make a new one to improve the clarity of Figure 1 in the updated draft.
>
> **Q5: The impact of CO2 using a vanilla ResNet backbone (without the momentum encoder).**
>
> Thanks for the suggestion! We show in **G3** in our general response a case without the momentum encoder.
>
> **Q6: Using the letter P both for the distribution and for the pseudo label.**
>
> Thank you for the suggestion.
> - When using CO2 with MoCo, since the gradients of the positive are stopped with the momentum encoder, P the positive-negative distribution *is* the pseudo-label for Q the query-negative distribution.
> - When using CO2 without the momentum encoder, P and Q are symmetric and are pseudo-labels to each other in the proposed symmetric KL objective. In this case, it is indeed not appropriate to state the “pseudo-label P”.
> - We favor the framework without the momentum encoder to present our idea as we believe it is more general. We will rephrase the usage of “pseudo-label P” in the updated draft.
>
> We would be more than happy to discuss any further questions!
>
> [1] Xinlei Chen, Haoqi Fan, Ross Girshick, and Kaiming He. Improved baselines with momentum contrastive learning. *Technical report*, arXiv:2003.04297.

---

### Official Review · AnonReviewer1 · 2020-10-28
**Review #1**

**Rating:** 5
**Confidence:** 5

**Review:**


<Summary>

This paper addresses the problem of unsupervised contrastive learning for visual representation. Its key idea is to use a consistency regularization method to resolve the issue of one-hot labels for instance discrimination on which most previous work have relied. The proposed CO2 method is implemented on top of MoCo and MoCo v2 and improves their representation performance on multiple classification and detection tasks.

<Strengths>

1. It borrows  a simple consistency regularize method from semi-supervised learning literature to tackle one issue of recent unsupervised contrastive learning for visual representation – one-hot label that cannot discriminate the semantic closeness to a query between negative samples.

2. The proposed approach is applied to recent SOTA MoCo methods and further improves their performance on image classification, detection and semantic segmentation tasks.

<Weakness>

1. Although this paper can clearly alleviate one issue of recent practice of contrastive learning, the technical novelty is limited.

(1) This paper proposes a new use of an existing technique (consistence regularization) to a new problem (unsupervised contrastive learning). Given that the technique is basic and well-known in semi-supervised learning literature, the proposal of its simple use (with no extension) bears little technical novelty.

(2) In my opinion, the proposal could be a good practice for unsupervised contrastive learning, but its contribution may not be sufficient enough to make this work as a legitimate full paper on ICLR as one of the top premier ML conferences.

2. Experimental evaluation should be improved.

(1) The proposed approach shows the SOTA performance on multiple computer vision tasks, but it largely attributes to the strong performance of MOCO variants.

(2) The performance gain of the proposed method over MOCO v2 is rather marginal as shown in Table 2 and 3.

(3) The proposed CO2 is only implemented on MOCO variants. In order to show the generality of the proposed method, it should be tested with multiple contrastive learning methods and show whether it can consistently improve the methods.

<Conclusion>

My initial decision is ‘reject’ mainly because the contribution is somewhat limited and more empirical justification for the method is required.

---

> ### Author Response · Authors · 2020-11-19
> **Response to Reviewer 1**
>
> We thank Reviewer 1 for the valuable comments. Below we discuss the points you have raised in detail.
>
> **Q1: The technical novelty is limited.**
>
> Thank you for appreciating the effectiveness of our work and taking it as a good practice for unsupervised contrastive learning. We discuss our technical novelty and our contributions to bringing a new perspective to unsupervised contrastive learning in **G1** in our general response.
>
> **Q2: The performance gain of the proposed method over MOCO v2 is rather marginal.**
>
> Thank you for bringing up this concern. We discuss this in **G4** in our general response.
>
> **Q3: The generality of the proposed method.**
>
> We agree that testing CO2 on multiple contrastive frameworks can better support the generality of our proposed method. We show CO2 is also effective with SimCLR in **G3** in our general response.
>
> We would be more than happy to discuss any further questions!

---

### Official Review · AnonReviewer3 · 2020-11-02
**incremental improvement to MoCo**

**Rating:** 6
**Confidence:** 4

**Review:**

This paper proposes to add a new consistency loss term to the momentum contrast (MoCo) framework for self-supervised visual representation learning.  A common strategy for self-supervised learning, employed by MoCo as well as others, is to learn invariance to a class of transforms.  Here, a deep network is trained on an instance discrimination task: among distractor images and a transformed (e.g. via data augmentation) version of the input (query) image, correctly identify the transformed example (a classification problem).

Consistency regularization formulates an alternate objective: learn a similarity function between images and, given a gallery of images, encourage the input query and positive example (transformed variant) to have a similar distribution of similarity over the gallery.  The similarity distribution can be treated as a pseudo-label.

The paper trains a variant of MoCo that combines the standard instance discrimination loss with this consistency regularization loss.  For optimal hyperparameters of this combination, experiments show small but consistent accuracy gains over the baseline MoCo in multiple scenarios: classification and semi-supervised learning on ImageNet, as well as classification, object detection, and semantic segmentation on PASCAL.

In terms of overall impact, the contribution of this paper appears to be an incremental improvement to the current self-supervised learning paradigm.  The consistency loss itself is not a new idea, as the paper cites Sajjadi et al. (2016).  In fact, Sajjadi et al. examine consistency loss in the context of self-supervised learning, which appears to further limit the novelty of the current paper to the specific contribution of doing so with MoCo.

As another limitation, the overall gains are small and it is not clear that they necessarily make MoCo+CO2 the top method.  For example, on ImageNet classification, SimCLR outperforms MoCov2+CO2 (69.3 to 68.0 in Table 1), though SimCLR is trained for more epochs.

---

> ### Author Response · Authors · 2020-11-19
> **Response to Reviewer 3**
>
> We thank Reviewer 3 for the valuable comments. Below we discuss the points you have raised in detail.
>
> **Q1: Incremental improvement to the current self-supervised learning paradigm.**
>
> Thank you for bringing up this concern. Please see **G1** in our general response.
>
> **Q2: The consistency loss itself is not a new idea, as the paper cites Sajjadi et al. (2016).**
>
> Thanks for discussing the relationship between our work and Sajjadi et al. (2016). We carefully checked this paper and believe it has *not* examined consistency loss for *self-supervised learning* (we take self-supervised learning as a form of *unsupervised learning* following [1]), but for *semi-supervised learning* and *fully-supervised learning*. In other words, some or all of their training samples are labeled, while our method applies to scenarios where all training examples are unlabeled.
>
> We agree there is a close relationship between our work and Sajjadi et al. (2016). Their usage of supervised loss is like instance discrimination loss in the context of our work, and their usage of unsupervised consistency loss is like our CO2 loss. We believe it is our contribution to point out the connection between current unsupervised contrastive frameworks and semi-supervised learning.
>
> **Q3: The overall gains are small and it is not clear that they necessarily make MoCo+CO2 the top method.**
>
> Thanks for bringing up this concern, and we discuss this in **G4** in our general response.
>
>
> We would be more than happy to discuss any further questions!
>
> [1] Kaiming He, Haoqi Fan, Yuxin Wu, Saining Xie, and Ross Girshick. Momentum contrast for unsupervised visual representation learning. In *CVPR*, 2020.

---

### Author Response · Authors · 2020-11-19
**General Response**

We thank all reviewers for their valuable comments and insightful suggestions! Below we address four common concerns in detail.

**G1: Concerns on novelty and the relation to semi-supervised learning.**
- We agree that consistency regularization is well-established in semi-supervised learning. However, we believe “a new use of an existing technique to a new problem” (R3) can be impactful as long as the new use is effective and, more importantly, brings a new perspective.
- We show that consistency regularization, though simple and low-cost, addresses the issue of uninformative one-hot labels used by popular contrastive frameworks well.
- The extension of consistency loss on contrastive frameworks is nontrivial. In semi-supervised learning, consistency loss on unlabeled samples is always combined with a supervised loss on labeled samples. However, in our work, there are no supervised labels. We show a surprising example that a model learned completely without human annotations can also be regularized by the consistency assumption.
- Our work brings a new perspective of unsupervised visual representation learning. It challenges the stereotype that the pretext task can only be self-*supervised* which constructs artificial labels for every sample (eg., a specific degree of rotation [1], a specific positive crop in instance discrimination [2]). In contrast, the pretext task can also be self-*semi-supervised*, allowing the task itself to be partially labeled when information for label construction is not enough and imposing a label is harmful.
- We believe our method is novel, “simple yet effective” (R4) and “likely opens new research directions” (R2).

**G2: Is symmetric KL necessary?**
- Symmetric KL is not necessary. Forward KL only can work too, but symmetric KL is conceptually and empirically better.
- Conceptually, the query crop and the positive crop are symmetric to each other in the end-to-end setting (e.g. SimCLR, with a huge batch size and without the momentum encoder). In this case, our proposed objective, symmetric KL, is natural. Given limited batch size, MoCo approximates the large size of negative crops with a momentum updated queue without altering the optimization objective. So we keep using symmetric KL for this approximation.
- Empirically, we experiment with forward ($\text{KL}(P||Q)$), reverse ($\text{KL}(Q||P)$) and symmetric KL, find all of them improve, and symmetric KL improves a bit more. On ImageNet-100 with ResNet18, we test different loss functions and report the linear classifier accuracy as below:

|Loss|Accuracy(%)|
|---|---|
|MoCo v1| 63.6 |
|MoCo v1 + Reverse KL|65.1|
|MoCo v1 + Forward KL|69.6$\pm$0.27|
|MoCo v1 + Symmetric KL|69.7$\pm$0.41|

We perform 3 runs and report standard deviation for forward and symmetric KL as their results are quite close. Forward KL performs on par with symmetric KL. Reverse KL alone can also bring gains. On ImageNet with ResNet50, forward KL only can improve linear accuracy of MoCo v1 from 60.6% to 63.2%, a bit less than 63.5% of symmetric KL.

**G3: Is CO2 effective without the momentum encoder?**
- We agree that SimCLR results on ImageNet with huge batches and extra long training can better support our claim, but we cannot afford it.
- We do believe that MoCo optimizes a similar objective as SimCLR. The difference lies in that MoCo approximates the large size of negative samples with a small batch size by a momentum updated queue.
- CO2 is a novel objective which we believe is independent of whether to use a momentum encoder.
- Empirically, we prove this point by applying CO2 to SimCLR on ImageNet-100 with ResNet18 (with batch size 128 and temperature 0.1) evaluated by linear classification accuracy:

|Method|Accuracy(%)|
|---|---|
|SimCLR|68.8|
|SimCLR + CO2|72.4|

As shown in the table, CO2 improves SimCLR by 3.5%. CO2 uses $\alpha$ 1.0 and $\tau_{con}$ 0.07.

**G4: Incremental performance on MoCo v2.**
- Our improvements on MoCo v2 are consistent through all evaluated scenarios including linear classification, semi-supervised learning, PASCAL classification, detection and segmentation. This shows that CO2 indeed provides a better visual representation. Moreover, our experiments on ImageNet-100, in a low-data regime, show larger margin over baseline contrastive frameworks, with and without a momentum encoder.
- MoCo and SimCLR are sensitive to the choice of hyper-parameters [3,4], and the hyper-parameters depend heavily on dataset properties [4]. Due to resource constraints, we tune our hyper-parameters on ImageNet-100, which might not transfer well to ImageNet. But still, CO2 shows consistent improvements on various tasks.
- MoCo and SimCLR are trained for 800/1000 epochs. SimCLR requires a huge batch size of 4096. Both are impossible for us, so we compare only with the setting of 200 epochs on MoCo (v2).

We will update our draft with the above materials. We would be more than happy to discuss any further questions!

---

> ### Author Response · Authors · 2020-11-19
> **Reference**
>
> [1] Gidaris, Spyros, Praveer Singh, and Nikos Komodakis. Unsupervised Representation Learning by Predicting Image Rotations. In *ICLR*, 2018.
>
> [2] Zhirong Wu, Yuanjun Xiong, Stella X Yu, and Dahua Lin. Unsupervised feature learning via nonparametric instance discrimination. In *CVPR*, 2018.
>
> [3] Xinlei Chen, Haoqi Fan, Ross Girshick, and Kaiming He. Improved baselines with momentum contrastive learning. *Technical report*, arXiv:2003.04297.
>
> [4] Ting Chen, Simon Kornblith, Mohammad Norouzi, and Geoffrey Hinton. A simple framework for contrastive learning of visual representations. In *ICML*, 2020.

---

### Author Response · Authors · 2020-11-24
**Summary of the Revision**

We thank all reviewers for the valuable and inspiring comments, which helped us improve our work. We have uploaded an updated draft. Major modifications are colored in red. Below is a summary of major updates:

1. Refine our contributions to bring a new perspective in Sec. 1 Introduction. [**R1**, **R3**]
2. Update Figure 1 with more explanations, axis labels and ticks, and increased size of arrowheads. [**R2**]
3. Provide intuition and justification of using symmetric KL in Sec. 2 Method. [**R2**, **R4**]
4. Refrain from using pseudo label P in Sec. 2 Method. [**R2**]
5. Update Table 2 with the mean of 3 runs. [**R2**]
6. Discuss the small transfer learning improvements, and include reference results of more state-of-the-art unsupervised learning methods in Sec. 3.4 Transfer Learning. [**All reviewers**]
7. Study the impact of CO2 on SimCLR in Sec. 3.5 Analysis. [**R1**, **R2**, **R4**]
8. Study the different choices of the consistency loss in Sec. 3.5 Analysis. [**R2**, **R4**]
9. Discuss the relation to consistency regularization in semi-supervised learning in Sec. 4 Related Work. [**R1**, **R3**, **R4**]

We hope that our response and revision address your concerns and questions. We are happy to provide discussion if you have any further concerns or comments.

---

### Decision · Program_Chairs · 2021-01-07
**Final Decision**

**Decision:**

Accept (Poster)

**Comment:**

This paper presents a simple yet effective approach to improve self-supervised contrastive approaches like MoCo. There are concerns with respect to novelty/simplicity and low improvements over MoCov2. AC believes that simplicity is good and while gains might not be as huge, they still show usefulness of new loss. It might also provide insights for future papers on self-supervised learning. Overall, the sentiment is that paper is above the bar.